# The Prognostic Value of the Surface Electromyographic Assessment of Pelvic Floor Muscles in Women with Stress Urinary Incontinence

**DOI:** 10.3390/jcm9061967

**Published:** 2020-06-23

**Authors:** Kuba Ptaszkowski, Bartosz Malkiewicz, Romuald Zdrojowy, Malgorzata Paprocka-Borowicz, Lucyna Ptaszkowska

**Affiliations:** 1Department of Clinical Biomechanics and Physiotherapy in Motor System Disorders, Faculty of Health Science, Wroclaw Medical University, Grunwaldzka 2, 50-355 Wroclaw, Poland; kuba.ptaszkowski@umed.wrocl.pl (K.P.); malgorzata.paprocka-borowicz@umed.wroc.pl (M.P.-B.); 2Clinic of Urology and Urological Oncology, Wroclaw Medical University, Borowska 213, 50-556 Wroclaw, Poland; romuald.zdrojowy@umed.wroc.pl; 3Department of Physiotherapy, Opole Medical School, Katowicka 68, 45-060 Opole, Poland; ptaszkowska.l@gmail.com

**Keywords:** pelvic floor muscle, surface electromyography, urinary incontinence

## Abstract

Background: The use of surface electromyography (sEMG) measurements to evaluate the bioelectrical activity of the pelvic floor muscle (PFM) during therapeutic intervention is now well established. This study investigates the diagnostic possibilities of sEMG in women with stress urinary incontinence (SUI). The aim of this study was to carry out objective assessments of the bioelectrical activity of the PFM in women after menopause and determine the prognostic value of sEMG for assessing the PFM in patients with SUI. Methods: This was a prospective, observational study that evaluated the bioelectrical activity of the PFM in postmenopausal women with or without SUI (SUI group, *n* = 89 vs. non-SUI group, *n* = 62). The study was carried out between January 2013 and December 2018 at the Clinic of Urology (Wroclaw, Poland). The protocol for all sEMG measurements of PFM activity consisted of following elements: “baseline”, “quick flicks”, “contractions”, “static hold”, and “rest tone”; we then compared these results between groups. To determine the optimal cutoff level for sEMG activation of the PFM to detect the occurrence of SUI, we performed receiver operating characteristic (ROC) curve analysis (with Youden’s index). Results: Significantly lower results were obtained for all PFM measurements in women with SUI. The optimum diagnostic cutoff for “baseline” was 3.7 μV (area under curve (AUC), 0.63), “quick flicks” was 9.15 μV (AUC, 0.84), “contractions” was 11.33 μV (AUC, 0.80), “static hold” was 9.94 μV (AUC, 0.84), and “rest” was 3.89 μV (AUC, 0.63). Conclusions: Measuring sEMG activity in the PFM may be a useful diagnostic tool to confirm the absence of SUI. We can expect that the sEMG activity of subjects with SUI will be lower than that of healthy people. In order to determine appropriate reference values for assessing sEMG activity data in the PFM, it is now necessary to conduct multicenter studies.

## 1. Introduction

Previous literature highlights the validity of using surface electromyography (sEMG) measurements when assessing pelvic floor muscle (PFM) bioelectrical activity with regard to therapeutic progress [1,2,3,4,5,6]. This assessment method is considered as an objective, noninvasive, and safe tool. The sEMG method is widely used in everyday physiotherapeutic practice as a tool for providing feedback from muscles [2,7,8,9,10]. This study may define the importance of sEMG in the diagnosis of urinary incontinence and could therefore provide a better understanding of the causes of these symptoms. The primary aim of this study was the objective assessment of bioelectrical activity of the PFM in women after menopause and the comparison of these values between women with and without symptoms of stress urinary incontinence (SUI). A secondary aim was to determine the prognostic value of sEMG for assessing the PFM in patients with SUI. We hypothesized that lower bioelectrical activity would be observed in people with SUI and that it would be possible to indicate values that will relate directly to these symptoms. Our final aim was to investigate the impact of bioelectrical activity on the absence of SUI in a multifactorial model.

## 2. Methods

This was a prospective, observational study that evaluated the sEMG activity of the PFM in postmenopausal women with or without SUI. The study protocol was approved by the Bioethics Committee of Wroclaw Medical University (KB-611/2012), and the study was conducted in accordance with the ethical principles of the Declaration of Helsinki. All patients provided written and informed consent.

This study included women in their postmenopausal period with or without SUI and was carried out between January 2013 and December 2018 at the Department of Urology, University Hospital in Wroclaw (Poland). We recruited female patients from the department and volunteers who responded to a notice published in the local media. All recruited women were subjected to an evaluation that assessed specific inclusion and exclusion criteria. The exclusion criteria were as follows: aged over 75 years; no history of menopause; gynecological surgeries; surgeries in the abdomen/pelvis over the previous 10 years; injuries of the lower extremities, pelvis, or spine that were evident on the day of examination; prolapse of the reproductive organs; and third-degree incontinence [11,12]. Patients were also excluded if they were experiencing infection or menstruation, had allergies to nickel, experienced pain during the examination, or refused to participate during the examination.

Study participants were divided into two groups depending on the occurrence of urinary incontinence symptoms (SUI group vs. non-SUI group). The occurrence, or absence, of SUI symptoms was evaluated by an experienced urologist. In the urological examination, the urodynamic testing was included in order to determine the occurrence of urinary incontinence, its type, and its degree, and an additional International Consultation on Incontinence Questionnaire—Short Form (ICIQ-SF) was also used. The basis for qualifying for the SUI group was the involuntary loss of urine at least once a week for the preceding 3 months.

The participants visited the clinic once at the beginning of the study, at which point a detailed medical history was collected. Prior to commencing the trial involving the use of sEMG to assess PFM performance, we assessed the correct action of the PFM during a contraction. Measurements were made in a supine position with bent lower limbs (with the feet placed on a medical couch and the lower limbs slightly abducted).

Bioelectrical activity of PFM was assessed by surface electromyography and a compatible endovaginal electrode. The protocol of all measurements of PFM activity consisted of the assessment of the following elements: “baseline” (10 s PFM activity at rest before functional measurements), “quick flicks” (10 s measurement, in which participants performed short, quick contractions of the PFM), “contractions” (5 × 10 s contractions, in which the participant tried to contract the PFM and hold for 10 s), “static hold” (in which the participant attempted to hold the PFM contraction for 60 s), and “rest tone” (10 s PFM at rest after functional measurements).

sEMG measurements were acquired using a MyoSystem 1400L eight-channel electromyographic apparatus (Noraxon, Scottsdale, AZ, USA). A Life-Care Vaginal Probe PR-02 (Everyway Medical Instruments, New Taipei City, Taiwan) vaginal probe was used to record the sEMG signal from the PFM. The pear-shaped probe has a total length of 7.6 cm and a maximal circumference of 2.8 cm. The length of the recording plate is 4.5 cm, and the active surface area is 7.68 cm^2^/band (stainless steel). During measurements, the surfaces were directed to the right and left sides to record PFM activity with minimal crosstalk during tasks. This probe was inserted up to the handle at the introitus of the vagina. The monopolar reference electrode was placed on the anterior superior iliac spine.

The electromyographic signal (EMG) signal was subjected to standard post hoc processing. First, data were rectified and smoothed using the root mean square (RMS) algorithm and then subjected to a filtering method to reduce the phase shift. A narrow band filter was used in the range from 50 to 1000 Hz (finite impulse response filter). The bioelectric activity of the muscles tested was expressed in microvolts.

We paid particular attention to the following pitfalls related to sEMG measurements: (1) attention was paid to the exact, repeatable position during measurements; (2) we imposed an absolute ban on patient movement during the examination period; (3) the probe placement was carefully checked and performed in the same way for each participant; (4) the same type of probe was used for each examination; (5) in order to exclude additional pressure on the pelvic floor structures, the examination was performed with an empty bladder, and we also ensured that the patient had defecated and was not bloated on the day of examination. In addition, the patient was taught not to activate the abdominal press during PFM activity.

The estimated sample size was calculated in accordance with a pilot study [11]. The means and standard deviations of functional PFM activity in the SUI group and the non-SUI group were used to estimate the sample size for a two-sample, unpaired-means *t*-test. We determined the following parameters: SUI group mean 9.4 μV; non-SUI group mean 14.2 μV; standard deviation 7.7 μV; null difference 0 μV. The alpha level was set at 0.05, and the power of the test was set to 0.9. On the basis of these parameters, the estimated sample size was 56 women for each group. In addition, we assumed that 10% of patients would withdraw from the study. The final sample size was defined as 62 participants per group.

Statistical analysis was conducted using Statistica 13 (TIBCO Software Inc., Palo Alto, USA). In measurements where the study participant performed the test more than three times, the reliability and repeatability of these measurements for sEMG activity was assessed using an intraclass correlation coefficient (ICC). In each case, *r* ≥ 0.90). The distributions of baseline characteristics in the SUI group and the non-SUI group were compared with the chi-squared test for proportions and either the *t*-test or the Mann–Whitney U-test for continuous variables. Univariable and multivariable logistic regression were used to evaluate the influence of individual predictor variables, as well as their combined effect in predicting the absence of SUI. Variables in the univariate analysis that indicated a relationship to outcome (*p* ≤ 0.30) were selected for the multivariate logistic regression model (variables without multicollinearity). In cases where several variables had a correlation coefficient >0.6, a maximum of two of such variables were selected for the model (the rotation method: Varimax with Kaiser normalization). Statistically significant predictors were subjected to a backward, stepwise logistic regression analysis to determine which combination of predictors best explained the response. To determine the optimal cutoff level for sEMG activation of the PFM to detect the occurrence of SUI, we performed receiver operating characteristic (ROC) curve analysis (with Youden’s index). Based on the Youden index, the variable sEMG activities in the PFM were categorized (low vs. high) and included in a new logistic regression model. For all comparisons, we defined α as 0.05.

## 3. Results

### 3.1. Comparison of the Demographic and Clinical Characteristics

In total, 222 women were recruited and screened for this study; 80 women were excluded from the study for various reasons (Figure 1). Finally, 142 women were enrolled in the study: 89 in the SUI group and 62 in the non-SUI group. There were no statistical differences between the SUI and non-SUI groups in terms of characteristics (Table 1).

Analysis showed that participants from the SUI and non-SUI groups presented with different sEMG values in all of the tests performed. Significant differences between the two groups were observed for “quick flicks”, which was significantly lower in women with SUI (on average 7.3 μV lower; *p* < 0.001). Significantly lower results were also noted in the assessment of sEMG activity during the “contraction” test (on average 6.6 μV lower; *p* < 0.001), during the “static hold” test (on average 6.8 μV lower; *p* < 0.001), at “baseline” (on average 6.8 μV lower; *p* = 0.010), and at “rest”(on average 0.6 μV lower; *p* < 0.012) in women with SUI (Table 2).

### 3.2. Assessment of the Impact of Variables on Stress Urinary Incontinence

The relationships between the absence of SUI and the sEMG activity of the PFM (continuous variables) and other variables (including age, body weight, height, body mass index (BMI), type of work, and the number of births) were determined by logistic regression analysis (Table 3). The regression analysis was performed in a study population in which the absence of SUI was related to higher sEMG activity in the PFM. In the multifactorial model, the variables that showed the most significant relationship were “quick flicks” (odds ratio (OR) = 1.34, *p* < 0.001) and BMI (OR = 1.34, *p* < 0.001) (Hosmer–Lemeshow test = 11.3, *p* = 0.19).

### 3.3. Diagnostic Capability of the sEMG

ROC curves were constructed separately for all sEMG tests (Figure 2) in order to present the diagnostic ability of these variables to predict SUI. The optimum diagnostic cutoff for “baseline” was 3.7 (area under the curve (AUC), 0.63; 95% confidence interval (CI), 0.53–0.72; sensitivity at 84% specificity, 36%), “quick flicks” was 9.15 (AUC, 0.84; 95% CI, 0.78–0.91; sensitivity at 60% specificity, 94%), “contractions” was 11.33 (AUC, 0.80; 95% CI, 0.73–0.87; sensitivity at 60% specificity, 90%), “static hold” was 9.94 [AUC, 0.84; 95% CI, 0.78–0.91; sensitivity at 70% specificity, 94%), and “rest” was 3.89 (AUC, 0.63; 95% CI, 0.54–0.72; sensitivity at 64% specificity, 58%).

Additionally, a study was conducted on a prospectively assessed group of women (*n* = 30) aged from 58 to 78 years (x = 66.9 years; SD = 6.7 years) to validate the main results. On the basis of the results obtained, SUI was found in 17 participants. AUC results were compared between the study and validation groups. No statistically significant differences were found between the results. In the validation group the results were as follows: “baseline”—AUC, 0.62; 95% CI, 0.52–0.72; “quick flicks”—AUC, 0.84; 95% CI, 0.77–0.91; “contractions”—AUC, 0.79; 95% CI, 0.71–0.87; “static hold”—AUC, 0.85; 95% CI, 0.78–0.90; and “rest”—AUC, 0.63; 95% CI, 0.53–0.73.

Next, the sEMG results were categorized according to determined cutoff points. Logistic regression analysis was reperformed (modeling the probability of the absence of SUI), taking into account binary variables for sEMG (high levels that were equal to or higher than the determined cutoff point vs. low levels that were lower than the determined cutoff point).

Regression analysis showed that the absence of SUI was related to higher levels of sEMG in the PFM. The multifactorial model showed that the variable with the most significant relationship was “static hold” (OR = 15.37, *p* < 0.001). In addition, the model also featured “quick flicks” (OR = 3.40, *p* = 0.10) (Hosmer–Lemeshow test = 0.15, *p* = 0.70) (Table 4).

## 4. Discussion

We observed statistically significantly lower sEMG activities in the PFM of women with SUI. We also observed a higher odds ratio for the absence of SUI symptoms in women who obtained results above the cutoff point (mainly considering results derived from the “quick flicks”, “contractions”, and “static hold” tests).

Differences in the sEMG activities of the PFM between the SUI group and the non-SUI group represent important information in the context of SUI diagnostics. For example, Koenig et al. [12] evaluated the activity of PFM in SUI patients and healthy women. Although these researchers did not directly compare the results between the SUI group and the healthy group, we noted that the sEMG of PFM activity was lower in the SUI group (resting activity: 8.9 μV vs. 19.6 μV; mean of maximum voluntary contraction (MVC): 26.7 μV vs. 70.1 μV). In the article by Aukee et al. [2], we also noted that there were differences in this activity; the mean functional PFM activity in the SUI group was 17 μV, while that in the healthy group was 19.5 μV. These differences were much smaller than those reported by Koenig et al. [12], and, as the authors themselves emphasize, the results from the healthy and incontinent women overlap, although patients with SUI had weaker EMG activity in 5-s contractions. Significantly lower PFM activities were also reported by Yang et al. [13]. When measuring pretest and post-test resting baseline, the differences oscillated around 1 μV. When measuring functional activity (rapid, tonic, endurance contraction), the results differed by 2–3 μV. It should be emphasized that the research by Yang et al. [13] was based on women after childbirth. These researchers concluded that the assessment of sEMG in the PFM is a reliable measure to reflect the clinical PFM status and to evaluate postpartum SUI. In addition, the researchers noted that this represents a method that can effectively identify women with, or at risk of, postpartum SUI, and recommended the routine screening of all postpartum women seeking gynecological care. Rett et al. [14] also provided information relating to sEMG activity of the PFM in women with SUI; these values oscillated around 21 μV and were therefore higher than the figures presented in the current study.

It should be noted that the results in these earlier articles show higher levels of activity in the PFM than observed in the current study; however, the mean age of the participants in the earlier studies was lower than in the present study. A large number of reports emphasize the fact that PFM activity decreases with age [8,9,15,16]. Only one paper describes the PFM activity of women with SUI at a similar age (mean 66.1 ± 8.7 years) to those described in our study. Alves et al. [8] examined postmenopausal women with urogynecological complaints (stress, urgency, or mixed urinary incontinences) and obtained mean PFM results of 15.44 ± 8.22 μV.

The existing literature shows that sEMG activities in the PFM differ markedly [2,5,8,9,12,13,14,17,18,19,20,21,22,23,24]. It is possible that these differences relate to the tests being carried out with different equipment, with different electrodes (shape, size), or with the patient lying in different positions. These groups also differed in terms of age and condition. However, in our opinion, it is clearly evident that people with PFM dysfunction present with lower results than healthy people; our present data also confirm this viewpoint.

The present study was limited by the fact that it was carried out in a single center. Applying the same research protocol in other centers would allow us to include a larger number of participants. It would also give us the possibility of including a wider range of population groups and the ability to compare results across centers. These factors would allow us to improve the generalizability of this study. Our results did not incorporate normalized sEMG test values. This is because normalization for the PFM assessment can be difficult and inappropriate and because we wanted to compare our data with data derived from other studies, which also presented non-normalized values.

## 5. Conclusions

Our data suggest that measuring sEMG activity in the PFM may be a useful diagnostic tool to confirm the absence of SUI. We can expect that the sEMG activity of subjects with SUI will be lower than that of healthy people. In order to determine appropriate reference values for assessing sEMG activity data in the PFM, it is now necessary to conduct multicenter studies on a larger population, taking into account factors such as uniform research methodology, age, and the patient’s health status.

## Figures and Tables

**Figure 1 jcm-09-01967-f001:**
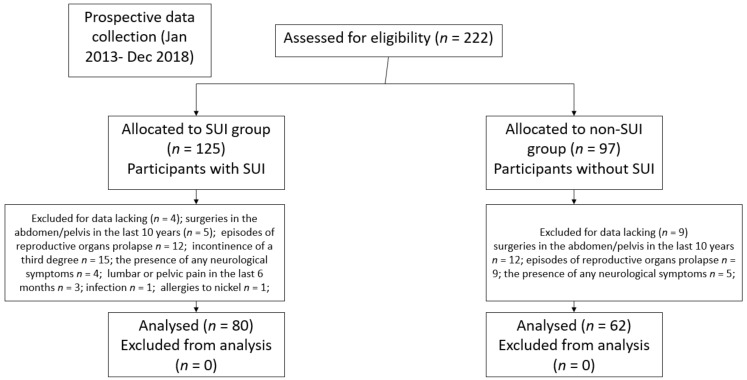
Flowchart showing the recruitment of participants (postmenopausal women, *n* = 220).

**Figure 2 jcm-09-01967-f002:**
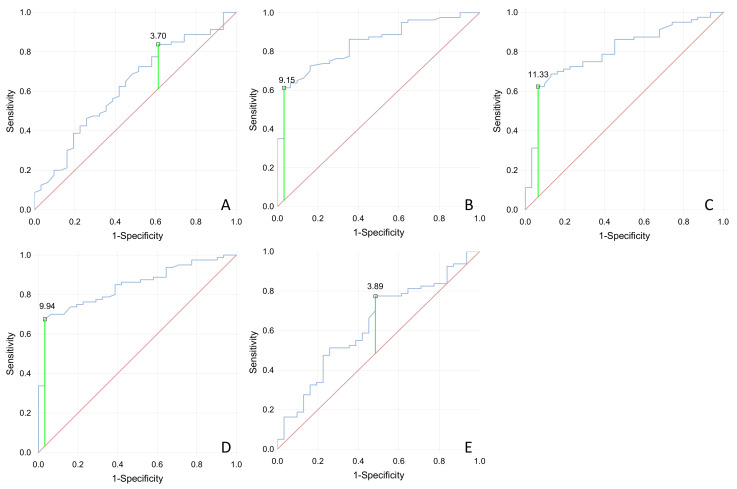
Receiver operating characteristic (ROC) curve (with cutoff point marked) for sEMG activity in the pelvic floor muscle of patients with SUI compared with non-SUI individuals. (**A**–**E**)—Baseline (**A**), Quick Flicks (**B**), Contractions (**C**), Static Hold (**D**), Rest (**E**).

**Table 1 jcm-09-01967-t001:** Comparison of the demographic and clinical characteristics of participants between the two groups.

Variables		SUI Group (*n* = 80)				Non-SUI Group (*n* = 62)				*p*-Value
		Mean	Min	Max	SD	Mean	Min	Max	SD	
**Age [years]**		63.5	45.0	78.0	7.9	63.2	46.0	79.0	9.1	0.23 *
**Body weight [kg]**		68.8	52.0	90.0	10.2	65.8	50.0	92.0	9.51	0.07 *
**Body height** **[cm]**		1.61	1.47	1.72	0.05	1.61	1.46	1.73	0.059	0.98 *
**BMI [kg/cm^2^]**		26.5	19.0	35.8	3.9	25.3	18.8	36.9	3.65	0.07 *
**ICIQ score**		8.7	0.0	19.0	4.4	0.0	0.0	0.0	-	-
**Type of work**	**Mental**	*n* = 84; 96%				*n* = 60; 96%				0.94 **
	**Physical**	*n* = 3; 4%				*n* = 2; 4%				
**Number of births**	**0**	*n* = 13; 15%				*n* = 4; 6.5%				0.31 **
	**1**	*n* = 23; 26%				*n* = 19; 31%				
	**2**	*n* = 37; 43%				*n* = 33; 53%				
	**3**	*n* = 11; 13%				*n* = 4; 6.5%				
	**4**	*n* = 3; 3%				*n* = 2; 3%				

ICIQ—International Consultation on Incontinence Questionnaire, Short Form; *n*—number of participants; Min—minimum value; Max—maximum value; SD—standard deviation; SUI—stress urinary incontinence; BMI—body mass index; * *t*-test; ** chi-squared test.

**Table 2 jcm-09-01967-t002:** Comparison of surface electromyography (sEMG) results between SUI and non-SUI women.

Variables	SUI Group (*n* = 80)				Non-SUI Group (*n* = 62)				*p*-Value *
	Mean	Min	Max	SD	Mean	Min	Max	SD	
**Baseline [μV]**	2.8	0.6	5.6	1.2	3.3	1.5	5.8	1.2	0.010
**Quick Flicks [μV]**	8.5	1.6	23.4	4.6	15.8	6.2	36.0	6.9	<0.001
**Contractions [μV]**	9.8	2.0	25.0	5.6	16.4	3.6	33.2	5.8	<0.001
**Static Hold [μV]**	8.6	2.0	28.3	5.1	15.4	5.3	31.7	5.6	<0.001
**Rest [μV]**	3.3	0.6	7.2	1.4	3.9	1.5	8.7	1.6	0.012

*n*—number of participants; Min—minimum value; Max—maximum value; SD—standard deviation. * *t*-test.

**Table 3 jcm-09-01967-t003:** Assessment of the impact of variables on the absence of stress urinary incontinence (univariate logistic regression analysis).

Variables		B	SE	*p*-Value	OR	95% CI Lower	95% CI Upper
**Age [years]**		0.00	0.02	=0.81	1.00	0.96	1.04
**Body weight [kg]**		−0.03	0.02	=0.07	0.97	0.94	1.00
**Body height** **[cm]**		−0.91	3.19	=0.78	0.40	0.00	209.48
**BMI [kg/m^2^]**		−0.08	0.05	=0.07	0.92	0.84	1.01
**Baseline [μV]**		0.38	0.15	=0.012	1.46	1.09	1.96
**Quick Flicks [μV]**		0.29	0.05	<0.001	1.34	1.21	1.49
**Contractions [μV]**		0.20	0.04	<0.001	1.22	1.13	1.32
**Static Hold [μV]**		0.26	0.05	<0.001	1.30	1.18	1.43
**Rest [μV]**		0.28	0.12	=0.015	1.33	1.06	1.67
**Type of work**	**Mental**	Ref.					
	**Physical**	−0.07	0.93	=0.94	0.93	0.15	5.76
**Number of births**	**0**	Ref.					
	**1**	0.99	0.65	=0.13	2.68	0.75	9.61
	**2**	1.06	0.62	=0.09	2.90	0.86	9.77
	**3**	0.17	0.82	=0.84	1.18	0.24	5.86
	**4**	0.77	1.08	=0.47	2.17	0.26	17.89

B—regression coefficient.; SE—standard error; OR—odds ratio; CI—confidence interval; BMI—body mass index.

**Table 4 jcm-09-01967-t004:** Assessment of the impact of sEMG activity (categorized variables) on the absence of SUI (univariate logistic regression analysis).

Variables	Level	SUI		B	SE	*p*-Value	OR	95% CI Lower	95% CI Upper
		Yes	No						
**Baseline**	**Low**	*n* = 67; 84%	*n* = 40; 65%	Ref.					
	**High**	*n* = 13; 16%	*n* = 22; 35%	1.04	0.40	0.010	2.83	1.29	6.24
**Quick Flicks**	**Low**	*n* = 48; 60%	*n* = 4; 6%	Ref.					
	**High**	*n* = 32; 40%	*n* = 58; 96%	3.08	0.57	<0.001	21.75	7.19	65.84
**Contractions**	**Low**	*n* = 50; 63%	*n* = 6; 10%	Ref.					
	**High**	*n* = 30; 38%	*n* = 56; 90%	2.74	0.49	<0.001	15.56	5.98	40.46
**Static Hold**	**Low**	*n* = 56; 70%	*n* = 4; 6%	Ref.					
	**High**	*n* = 24; 30%	*n* = 58; 94%	3.52	0.57	<0.001	33.83	11.03	103.74
**Rest**	**Low**	*n* = 51; 64%	*n* = 26; 42%	Ref.					
	**High**	*n* = 29; 36%	*n* = 36; 58%	0.89	0.35	0.010	2.44	1.23	4.81

B—regression coefficient; SE—standard error; OR = odds ratio, CI = confidence interval; SUI—stress urinary incontinence.

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
