# Peer review of "The Prognostic Value of the Surface Electromyographic Assessment of Pelvic Floor Muscles in Women with Stress Urinary Incontinence"

_jcm, 2020, doi:10.3390/jcm9061967_

Round 1

Reviewer 1 Report

JCM-830361

This is an interesting study on the prognostic value of the surface electromyographic assessment of pelvic floor muscles in women with stress urinary incontinence.

The paper is of high quality and I only have a few minor points to make:

Abstract, page 1; lines 22-24

“This study may demonstrate the diagnostic possibilities of sEMG in women with stress urinary incontinence (SUI).”

The “may” in this sentence renders it almost meaningless. I suggest redrafting, for example: “This study investigates the diagnostic possibilities of…”.

Page 2; line 70

How did you define “third degree incontinence”? Please add a reference.

Table 2

“Besline” should be “Baseline”

I recommend this paper for publication with these minor revisions.

Author Response

  1. Reviewer 1 (comments and questions):

Reviewers' comments: “This is an interesting study on the prognostic value of the surface electromyographic assessment of pelvic floor muscles in women with stress urinary incontinence. The paper is of high quality and I only have a few minor points to make:”

Authors' response:

  • Thank you very much for your comments.

Reviewers' comments: Abstract, page 1; lines 22-24

“This study may demonstrate the diagnostic possibilities of sEMG in women with stress urinary incontinence (SUI).”

The “may” in this sentence renders it almost meaningless. I suggest redrafting, for example: “This study investigates the diagnostic possibilities of…”.

Authors' response:

  • This sentence has been changed: This study investigates the diagnostic possibilities of sEMG in women with stress urinary incontinence (SUI)

Reviewers' comments: Page 2; line 70

How did you define “third degree incontinence”? Please add a reference.

Authors' response:

  • The third degree of incontinence was determined by the urologist who qualified to participate in the project. Urinary incontinence assessment was based on urodynamic examination. A detailed interpretation of the study can be found:

Ferrari A, Baresi L, Frigerio L, Costa M. A grading model for stress urinary incontinence. Urology. 1986;27(1):76‐78. doi:10.1016/0090-4295(86)90214-1

Mayekar RV, Bhosale AA, Kandhari KV, Nandanwar YS, Shaikh SS. A study of transobturator tape in stress urinary incontinence. Urology Annals. 2017 Jan-Mar;9(1):9-12. DOI: 10.4103/0974-7796.198867.

We have added these items to the references.

  • Also, we added this sentence (line 76-79):

In the urological examination, the urodynamic testing was included, in order to the occurrence of urinary incontinence, its type, and a degree was determined and an additional International Consultation on Incontinence Questionnaire - Short Form (ICIQ-SF) was used.

Reviewers' comments: Table 2

“Besline” should be “Baseline”

Authors' response:

  • This word has been corrected.

I recommend this paper for publication with these minor revisions.”

Authors' response:

  • Thank you.

Reviewer 2 Report

The authors describe the evaluation of the pelvic floor muscle in patients with stress urinary incontinence using surface electromyography (sEMG). A numbers of diagnostic measures (e.g. baseline, quick flicks etc) are assessed to determine optimal cut-offs for diagnostic application. 

Major issues: 

1) The categorisation of SUI/non-SUI groups is based on "ICIQ-SF questionnaire and involuntary loss of urine at least once a week for the preceding 3 months". This information would be insufficient to accurately diagnose of stress urinary incontinence. Conventionally, urodynamics is required for diagnosis or a combination of symptoms assessment (e.g. with cough stress test), voiding diary, PVR urine (<50mL). 

2) The authors evaluated a relatively large sample population (n=142) and conducted a number of sEMG measures based on pelvic floor muscle at rest or contracted. Whilst there is some novel information such variations of contractility tests there is little differentiation between this study and similar past studies defining sEMG results in stress urinary incontinent patients. Such as: 

Gunnarsson and Mattiasson (1999) Neurourology and urodynamics, 18(6), pp.613-621.

Decreased vaginal contractility measured by sEMG in women with stress incontinence compared to healthy. Repeat short (2s) squeezes where performed in 144 women and 58 with stress incontinence.  

Glazer et al. (1999) J Reprod Med, 44, pp.779-782.

Reduced contractility during pelvic floor muscle hold manoeuvrer (10s) showed significant differences between healthy and stress urinary incontinent patient (n=57).

Aukee et al (2003) Maturitas, 44(4), pp.253-257. 

sEMG activity in 31 stress incontinent and 35 healthy women assessed during rapid contractions (5s) in supine and standing. No significant differences in muscle activity in supine position, while significantly weaker sEMG in stress incontinence compared to healthy in standing position.  

3) The most effective diagnostic variables identified by the authors were "quick flick" AUC 0.84 and "static hold" AUC 0.84. Similar contractility measures (hold, rapid contraction etc.) have been utilised by past sEMG assessment of urinary incontinence. Authors should consider testing the accuracy of these variables in a new test group (separate cohort to the development group).    

Author Response

Reviewer 2.

Reviewers' comments: “he authors describe the evaluation of the pelvic floor muscle in patients with stress urinary incontinence using surface electromyography (sEMG). A numbers of diagnostic measures (e.g. baseline, quick flicks etc) are assessed to determine optimal cut-offs for diagnostic application.

Major issues:

1) The categorisation of SUI/non-SUI groups is based on "ICIQ-SF questionnaire and involuntary loss of urine at least once a week for the preceding 3 months". This information would be insufficient to accurately diagnose of stress urinary incontinence. Conventionally, urodynamics is required for diagnosis or a combination of symptoms assessment (e.g. with cough stress test), voiding diary, PVR urine (<50mL).

Authors' response:

  • Thank you for this comment. We agree. The classification of people into a given group was conducted by an experienced urologist (author B.M.). This information has been included in the text (Line:76-77): The occurrence, or absence, of SUI symptoms, was evaluated by an experienced urologist.
  • In the urological examination, the urodynamic testing was included, in order to the occurrence of urinary incontinence, its type, and a degree was determined. We added this sentence (line:77-79):

In the urological examination, the urodynamic testing was included, in order to the occurrence of urinary incontinence, its type, and a degree was determined and an additional International Consultation on Incontinence Questionnaire - Short Form (ICIQ-SF) was used.

2) The authors evaluated a relatively large sample population (n=142) and conducted a number of sEMG measures based on pelvic floor muscle at rest or contracted. Whilst there is some novel information such variations of contractility tests there is little differentiation between this study and similar past studies defining sEMG results in stress urinary incontinent patients. Such as:

Gunnarsson and Mattiasson (1999) Neurourology and urodynamics, 18(6), pp.613-621.

Decreased vaginal contractility measured by sEMG in women with stress incontinence compared to healthy. Repeat short (2s) squeezes where performed in 144 women and 58 with stress incontinence. 

Glazer et al. (1999) J Reprod Med, 44, pp.779-782.

Reduced contractility during pelvic floor muscle hold manoeuvrer (10s) showed significant differences between healthy and stress urinary incontinent patient (n=57).

Aukee et al (2003) Maturitas, 44(4), pp.253-257.

sEMG activity in 31 stress incontinent and 35 healthy women assessed during rapid contractions (5s) in supine and standing. No significant differences in muscle activity in supine position, while significantly weaker sEMG in stress incontinence compared to healthy in standing position. 

Authors' response:

  • Yes, these publications were in some sense an inspiration to carry out this study. They showed the difference between people with SUI and healthy people. However, no study attempted to determine the cut point in the results between these groups. In our research, the essence was an attempt to determine the cut point value in the results of the electromyographic examination. Based on this kind of study, it will be possible to exclude SUI in the future where weak pelvic floor muscles would be the prime cause.

3) The most effective diagnostic variables identified by the authors were "quick flick" AUC 0.84 and "static hold" AUC 0.84. Similar contractility measures (hold, rapid contraction etc.) have been utilised by past sEMG assessment of urinary incontinence. Authors should consider testing the accuracy of these variables in a new test group (separate cohort to the development group).   ”

Authors' response:

  • Thank you very much for this tip. Finally, our next step is to use our research to assess the reliability of this method based on research in a new population to assess the predictive effectiveness of this method. We can assure you that such work is ongoing.

Reviewer 3 Report

The manuscript in title of “The prognostic value of the surface electromyographic assessment of pelvic floor muscles  in women with stress urinary incontinence” aims to  assess the bioelectrical activity of the pelvic floor muscle (PFM) in women after menopause and determine the prognostic value of surface electromyography (sEMG) for assessing the PFM in patients with stress urinary incontinence (SUI). The authors claimed that measuring sEMG activity in the PFM may be a useful diagnostic tool to confirm the absence of SUI. This is very nice study! It has high potential for the diagnosis of SUI and pelvic floor dysfunction. However, several issues should be fixed before it could be accepted for publication.

  1. In the method section, more information should be provided for the sEMG measurement as the location is critical for sEMG.
  2. The resolution of Fig 1 and Fig 2 is too low, especially the words embedded. Please provide images >300 dpi.
  3. Results are not well organized, please think about to add subtitle in result section.

Author Response

Reviewer 3.

Reviewers' comments: “The manuscript in title of “The prognostic value of the surface electromyographic assessment of pelvic floor muscles  in women with stress urinary incontinence” aims to  assess the bioelectrical activity of the pelvic floor muscle (PFM) in women after menopause and determine the prognostic value of surface electromyography (sEMG) for assessing the PFM in patients with stress urinary incontinence (SUI). The authors claimed that measuring sEMG activity in the PFM may be a useful diagnostic tool to confirm the absence of SUI. This is very nice study! It has high potential for the diagnosis of SUI and pelvic floor dysfunction. However, several issues should be fixed before it could be accepted for publication.

In the method section, more information should be provided for the sEMG measurement as the location is critical for sEMG.

Authors' response:

  • Thank you for this comment. We have added the following information to the manuscript:

The pear-shaped probe has a total length of 7.6 cm and a maximal circumference of 2.8 cm. The length of the recording plate is 4.5 cm and an active surface area is 7.68 cm2/band (stainless steel). During measurements, the surfaces were directed to the right and left sides to record PFM activity with minimal crosstalk during tasks. This probe is inserted up to the handle at the introitus of the vagina. The monopolar, reference electrode was placed on the anterior superior iliac spine.

The resolution of Fig 1 and Fig 2 is too low, especially the words embedded. Please provide images >300 dpi.

Authors' response:

  • The resolution has been changed. Files saved in the new .emf format. The font has also been increased.

Results are not well organized, please think about to add subtitle in result section.”

Authors' response:

- Subtitles have been added to the results section:

  • Comparison of the demographic and clinical characteristics.
  • Assessment of the impact of variables on the stress urinary incontinence.
  • Diagnostic capability of the sEMG.

Round 2

Reviewer 2 Report

Thank you for clarifying the first issue, the added sentence will alleviate any confusion. 

Regarding the second issue, in this study the authors present sEMG mean, range and SD values for SUI patients and non-SUI volunteers. For example, Static Hold SUI = 8.6 (2-28.3) SD 5.1 and Non-SUI = 15.4 5.3-31.7 SD5.6The cut-point for static hold is 9.94 (sensitivity 60%; specificity 94%). 

Previous studies have provided mean and SD/range but, as the authors mentioned, potential cut-points have been provided in this study. These cut-points would serve to aid in the diagnosis of women with SUI. Though, using these cut-points to differentiate between patients with SUI and weak pelvic floor muscle is questionable as this study only compared SUI patients with non-SUI volunteers (cut-points for weak pelvic floor patients were not investigated).   

Given the primary differentiator between this study and previous work is the cut-points, it further highlights the need for their validation. The large sample size used in this study has produced a robust set of cut-points, however, this reviewer highly recommends a validation study on a small cohort to confirm cut-point reliability.

Another suggestion would be to test all or a selection of cut-points against retrospective clinical sEMG data from SUI and non-SUI patients. This should not require additional resources and can be conducted reliatively quickly but would significancy improve the validaty of this study.   

Author Response

Response to Reviewer:

Thank you for this comment. I fully agree with this recommendation. Due to the fact that the research is still ongoing, we randomly selected 30 new patient results. We conducted validation based on the AUC and ROC curves (AUC comparison)(James A. Hanley, Karim O. Hajian-Tilaki,
Sampling variability of nonparametric estimates of the areas under receiver operating characteristic curves: An update, Academic Radiology, Volume 4, Issue 1, 1997, Pages 49-58, ISSN 1076-6332, https://doi.org/10.1016/S1076-6332(97)80161-4.)

The results we have attached are described in the manuscript (line 185-191):

"Additionally, a study was conducted on a prospectively assessed group of women (n = 30) aged from 58 to 78 years (x = 66.9 years; SD = 6.7 years) to validate the main results. On the basis of the results obtained, SUI was found in 17 participants. AUC results were compared between the study and validation groups. No statistically significant differences were found between the results. In the validation group the results were as follows: "baseline" - AUC, 0.62; 95% CI, 0.52 - 0.72; "quick flicks" - AUC, 0.84; 95% CI, 0.77- 0.91, “contractions” - AUC, 0.79; 95% CI, 0.71 - 0.87, "static hold" - AUC, 0.85; 95% CI, 0.78 - 0.90, and "rest" - AUC, 0.63; 95% CI, 0.53 - 0.73."

Detailed results are included in the table in supplementary materials.